# DaHuangWan targets EGF signaling to inhibit the proliferation of hepatoma cells

Ha Si[1,2,3☯], Ba Genna[2☯], Xiangjin Zhuang[3‡], Jing Wang[3,4‡], B. Burenbatu[1]*, Qiyu Feng[3]*, Hongyang Wang[3]*

1 Affiliated Hospital of Inner Mongolia University for the Nationalities, Tongliao, Inner Mongolia, China, 2 Inner Mongolia University for the Nationalities, Tongliao, Inner Mongolia, China, 3 National Center for Liver Cancer, Eastern Hepatobiliary Surgery Hospital/Institute, the Second Military Medical University, Shanghai, China, 4 Division of Life Sciences and Medicine, Cancer Research Center, The First Affiliated Hospital of USTC, University of Science and Technology of China, Hefei, Anhui, China

☯ These authors contributed equally to this work.
‡ These authors also contributed equally to this work.
* qiyu@icloud.com (QF); hywangk@vip.sina.com (HW); brbt2013@163.com (BB)

**Data Availability Statement:** All relevant data are within the paper and its Supporting Information files.

**Funding:** This work was supported by grants from Natural Science Foundation of Inner Mongolia (NO.

## Abstract

DaHuangWan (DHW) is a traditional herbal medicine used by Mongolian to treat liver cancer for many years. Clinical application of the drug has been shown to help control tumor progression, prolong survival and improve quality of life. However, the underlying mechanisms and side effects of this drug remain unclear, which greatly limits the clinical application and further optimization of DHW. In this study, we found that DHW inhibits the proliferation of hepatoma cells by modulating the epithelial growth factor (EGF) signaling pathway. Berberine and Costunolide are the main active ingredients in DHW. Interestingly, the combination of Berberine and Costunolide has a dramatic synergistic effect on inhibiting the proliferation of hepatoma cells. Neither Berberine nor Costunolide directly block EGFR phosphorylation. Berberine promotes endocytosis of activated EGFR, while as Costunolide increases ubiquitination of EGFR and reduces EGFR recycling to cell membrane distribution, thereby inhibiting EGF signaling. Berberine and Costunolide target two different steps in regulating the EGF signaling, which explains the synergistic anti-cancer effect of DHW. Since Berberine and Costunolide do not directly target EGFR phosphorylation, DHW could be a supplementary medicine to tyrosine kinase inhibitors in cancer therapy.

## Introduction

Liver cancer is one of the most common and fatal malignant tumors, with 5-year survival rate estimated at 20% - 30%. According to GLOBOCAN, around 782,000 people were estimated to die of the disease in 2018, making it the fourth leading cause of cancer death worldwide [1]. In recent years, the incidence of primary liver cancer has risen dramatically, and hepatocellular carcinoma (HCC), the predominate pathological type of primary liver cancer, registered the highest and the most rapid rate of increase in the recent period of time [2]. Therefore, it is vital to find new drugs and strategies for HCC treatment.

2018MS08038, to HS) and National Natural
Science foundation of China (NSFC, 81773112, to
QF).

**Competing interests:** The authors declare no
potential conflicts of interest.

DaHuangWan (DHW), a traditional herbal formula, consists of two herbal ingredients,
Coptidis Rhizoma and Aucklandia lappa Decne, with a ratio of 1:1 (w/w) [3]. It has been used
clinically in Mongolia for many years to treat gastroenterological disorders and HCC. How-
ever, there are no well-controlled scientific experiments to verify the validity of DHW pre-
scription, and the underlying mechanism of the drug remains unclear, which greatly limits the
clinical application and further optimization of DHW.

Traditional herbs are usually used in combination. Herb pair, a basic unit in multi-herbal
formula, consists of two single herbs and usually has better pharmacological effects than using
these herbs alone [4]. Coptidis Rhizoma (HuangLian) and Aucklandia lappa Decne are com-
monly used as paired herbal medicine for cancer treatment. According to chinese traditional
medical literature, Coptidis Rhizoma is a widely used traditional chinese herb that eliminates
'heat' and 'toxicity'. The anti-cancer effects of Coptidis Rhizoma may be due to these tradi-
tional medical properties of Coptidis Rhizoma [5]. It has been reported that Coptidis Rhizoma
extract and its active ingredient, Berberine, has an anti-tumor effect on various human cancers
[5]. Coptidis Rhizoma extract inhibits the migration and invasion of HCC cells by down-regu-
lating the RHO/ ROCK signaling pathway [6]. Berberine induces death of human hepatoma
cells in vitro by down-regulating CD147 [7]. Aucklandia lappa Decne has been used in China
for the treatment of asthma, anorexia, nausea, ulcers and stomach problems for many years
[8,9]. It is also considered as an anti-cancer herb. Recent studies have shown that the ethanol
extract of Aucklandia lappa Decne has anti-cancer effect on prostate cancer, oral cancer, breast
cancer, and cervical cancer [10–12]. In addition to its anti-tumor chemo-preventive effect,
Costunolide also has anti-cancer activity against various cancer cells such as lung cancer,
breast cancer, and liver cancer, and can inhibit the invasion and metastasis of cancer cells [13–
19]. Although Berberine and Costunolide have decent anti-cancer activity, it is not clear
whether Berberine and Costunolide are key functional components of DHW, and the benefits
of this combination for liver cancer treatment. In this study, we examined the effect of DHW
on hepatoma cell proliferation and further explored its underlying mechanisms.

Epidermal growth factor (EGF) signaling is a core signaling pathway that regulates cell pro-
liferation. Epidermal growth factor receptor (EGFR) amplification and its abnormal activity
are tightly linked to the occurrence and development of various malignant tumors including
liver cancer [20,21]. Therefore, key molecules in EGFR signaling are considered to be impor-
tant oncogenic factors and critical therapeutic targets. For example, Cetuximab, a chimeric
(mouse/human) monoclonal antibody against EGFR, was approved by FDA in 2004 and by
CHMP in 2008 in combination with platinum-based therapy for the treatment of patients with
squamous cell carcinoma of the head and neck with metastatic disease, and in combination
with radiation therapy for locally advanced cancer [22]. Gefitinib, a small molecular EGFR
inhibitor, is approved for the treatment of patients with non-small cell lung cancer (NSCLC)
after failure of both platinum-based or docetaxel chemotherapies [22,23].

In addition to EGFR phosphorylation, homeostasis of EGFR is also critical for EGF signal-
ing. This homeostasis is maintained by modulating endocytosis, degradation and recycling of
the EGFR [24–26]. When EGFR is activated, endocytosis occurs through two pathways: cla-
thrin-dependent and clathrin-independent. Typically, EGFR binds to a ligand and is engulfed
by primary endocytosis, converting from a membrane distribution to an intracytoplasmic vesi-
cle distribution. After signaling to downstream molecules, a portion of the EGFR can escape
from primary endocytosis and be recycled to the membrane to continue binding to the ligand.
For EGFR that cannot be separated from primary endocytosis, the ubiquitin-protease system
will be activated to degrade EGFR when primary endocytic vesicles grow into secondary vesi-
cles and bind to lysosomes [27,28]. EGFR homeostasis is maintained by receptor internaliza-
tion processes, including endocytosis, ubiquitination and degradation, leading to well-

controlled activation of downstream signaling cascades. The ubiquitination of EGFR is a key regulatory node of this process and is then critical for the output of EGF signaling.

In the present study, we demonstrate that DHW inhibits the proliferation of hepatoma cells by modulating the EGF signaling pathway. In addition, the two active ingredients of DHW, Berberine and Costunolide have a dramatic synergistic effect on inhibiting the proliferation of liver cancer cells. This synergistic effect can be attributed in part to Berberine promoting endocytosis of activated EGFR, while Costunolide increases ubiquitination of EGFR and reduces EGFR recycling to cell membrane distribution.

## Materials and methods

### Chemicals and reagents

Cell cycle and apoptosis analysis kit, EdUrd Cell Proliferation Kit with Alexa Fluor 488 and Anti-ubiquitin antibody were purchased from Beyotime (Nanjing, China); Anti-EGFR, Anti Phospho-EGFR and β-actin antibodies were from Abcam(Cambridge, UK); The herb plant materials of Coptidis Rhizoma and Aucklandia lappa Decne were purchased from TongRen-Tang pharmacy (BeiJing, China); Berberine and Costunolide with the purity of 98.8% used as the positive control was obtained from food and drug testing institute (Bei Jing, China).

### Preparation of DHW extracts and determination of its active constituents

According to the proportion of the prescriptions of DHW, weighed the same amount of each medicinal herbs and crushed them. 500 g of the mixed drug powder was taken, and 8 times of 75% ethanol was added there to, and the mixture was heated under reflux for 3 times for 1.5 hrs, filtered, and the filtrate was combined. Ethanol was recovered by evaporation on a rotary evaporator and dried in a vacuum rotary concentrator.

The sample was analyzed using a Shimadzu Prominence LC-20A series HPLC apparatus (Shimadzu Co., Kyoto, Japan) equipped with auto sample injector (SIL-20AC) and PDA photo diode array detector (SPD-M20A). Berberine was separated by water and acetonitrile (50:50, v/v) at a flow rate of 1.0 ml/min. The column used for Berberine was the Hypersi1 ODS2 (100 mm × 4.6 mm, 5 μm particle size), and Berberine was detected at 345 nm. The mobile phase for Costunolide was comprised of acetonitrile/water (65:35, v/v). The solvent flow rate was 1.0 ml/min and Costunolide was detected at 225 nm. Contents of Berberine and Costunolide in DHW extracts were determined by referring to the calibration curve established by running standard at varying concentrations under the same conditions.

### Cell lines and cell culture

Human hepatocellular carcinoma cell lines (PLC/PRF/5, SMMC-7721, Hur-7, HCC-LM3 and HepG2) and QSG-7701 hepatocyte cell line were obtained from the Shang Hai Cell Biology Institute of Chinese Academy of Science (Shanghai, China). The cells were cultured in a medium containing 10% FBS and RPMI 1640, and changed every 2 days.

### Cell viability assay

All kinds of cancer cells were seeded in 96-well cell culture plates at the concentration of $2\times10^3$ ~$5\times10^3$ cells per well. After incubation overnight, the medium was removed and replaced with fresh medium with or without DHW extract. Cell density was measured on day 1, 2, and 3 by using the3-(4,5-dimethylthiazol-2yl)-2,5-diphenyltetrazolium bromide (MTT) following the manufacturer's instructions. The absorbance of converted dye is measured at the wavelength

of 490 nm and the absorbance is directly proportional to cell viability. All experiments were repeated at least three times.

## Analysis of cell cycle and apoptosis by flow cytometry

To determine the cell cycle, $2 \times 10^6$ cells/well were seeded in 6-well cell culture plates and treated with DWH at a final concentration of 40 $\mu$g/ml. After 48 hrs treatment, both floating and trypsinized adherent cells were collected and fixed with 70% ethanol. After fixation, the cells were washed with ice PBS and stained with propidium iodide (PI) for 30 mins under subdued light. Stained cells were analyzed using BD FACS Calibur and CellQuest software (BD LSRFortessa, USA). For cell apoptosis, PLC/PRF/5 cells were indicated, collected, and stained with Houchst33342 and PI as recommended by the manufacturer. Apoptotic cells were detected by flow cytometry, and the extent of apoptosis was calculated with FlowJo software (version 7.6.1).

## EdUrd incorporation assay

To determine EDUrd incorporation, PLC/PRF/5 cells were seeded in sterile chamber-well slides. After 24 hrs of plating, incubation was continued for another 48 hrs in absence (control) or presence of different testing agents as described in the legends to the figures. At the end of the 48 hrs incubation period, added to the medium along with EDUrd (20 μM). After 2 hrs, the percentage of EDUrd positive cells were determined examined a fluorescence microscope.

## Immunoprecipitation and Western blot analysis

After drug treatment, PLC/PRF/5 cells were lysed with NP-40 buffer and the protein concentration of was determined by Bradford protein assay kit (thermo fisher). For Immunoprecipitation, equal amount (0.2 ~ 1 mg) of total protein samples were incubated with EGFR antibodies at 4˚C overnight with gentle mixing, followed by the addition of 20 $\mu$l of protein A/G beads and continued incubation for another 2 hrs. The beads were pelleted via centrifugation at 2500 rpm for 3 mins and washed five times with NP-40 buffer. The precipitated proteins were eluted in 40 $\mu$l of lysate sample buffer. For Western blot analysis, the proteins were subjected to 4~15% SDS-PAGE and transferred to nitrocellulose membranes (Amersham Biosciences, Buckinghamshire, UK). The membranes were blocked with 5% milk and were incubated with the indicated primary antibodies. Odyssey infrared fluorescence scanning imaging system can read the film and conduct semi-quantitative analysis.

## Cell surface protein biotinylation and internalization assay

Cell surface protein was isolated using Pierce Cell Surface Protein Isolation Kit (Pierce, Rockford, IL). Briefly, after drug treatment, PLC/PRF/5 cells were washed three and suspended in ice-cold PBS (pH 8.0). Biotinylation was performed by incubating cells in PBS containing EZ-link Sulfo–NHS–SS–biotin for 30 mins. Cells were subsequently washed three times with ice-cold PBS to remove unreacted reagent. Labelled cells were lysed and cell surface proteins were immunoprecipitated with avidin resin and analyzed by immunoblot with EGFR antibody.

EGFR internalization was performed using Pierce Cell Surface Protein Isolation Kit as well. Briefly, the proteins on the cells surface were first Biotinylated with reducible Sulfo-NHS-SS-biotin at 4 ˚C. The biotin reaction was quenched for 10 mins with 0.1 M lysine (Thermo Fisher). After they were washed with PBS, cells were added to pre-warmed culture medium (without or with the different treatment compounds) and incubated at 37˚C for the indicated time. After stimulation with EGF 10ng/ml, cell surface biotin was removed by treatment (2 × 20 mins) at 4

˚C with sodium 2-mercaptoethanesulphonate (MesNa). The internalized proteins were immu-noprecipitated with avidin resin and analyzed by immunoblot with EGFR antibody.

## Statistical analysis

All statistical analyses were performed using Graph Pad Prism. The data shown represents the mean ± SD from at least three independent experiments. Student's t-test was used to compare two groups. ANOVA (One-way analysis of variance) used to analyze statistical differences between groups under different conditions. P values < 0.05 is considered to have significance.

## Result

### DHW extract inhibits the proliferation of hepatoma cells

DHW is a traditional herbal medicine used in Mongolia for the treatment of liver cancer for many years. The drug is claimed to have a clinically inhibitory effect on tumor progression. But so far, there are no well-controlled scientific experiments to verify the validity of DHW prescription. In this study, we first prepared DHW extract by refluxing DHW in ethanol. Then, we examined the effects of DHW extract on the growth of various liver cancer cells. PLC/PRF/5 hepatoma cells or QSG-7701 human hepatocytes were seeded in 12-well plates for 24 hrs and then these cells were treated with DHW extract. The number of cells was counted after 2, 4, and 6 days. We found that DHW extract significantly inhibited the growth of PLC/PRF/5 cell, while it had limited effect on the growth of QSG-7701 cells (Fig 1A and 1B). We then further quantified the inhibitory effect of DHW extract on various liver cancer cells. SMMC-7721, Hur7, HepG2, and LM3 cells were seeded in 96-well plates for 24 hrs and then treated with different dose of DHW extract (20 ~ 500 $\mu$g/ml). Cell viability was measured after 24, 48 and 72 hrs. The IC$_{50}$ values of these liver cancer cells was calculated (Table 1). As shown in Fig 1C, DHW extract inhibited the growth of SMMC-7721, HepG2, Hur7, and LM3 cells in a dose-and time-dependent manner, and these cells have different sensitivities against DHW extract. The effect of DHW on the QSG-7701 human hepatocytes was limited (Fig 1B), indicating that the inhibition of DHW on the growth of various liver cancer cell lines should not be derived from cytotoxicity.

To determine whether DHW inhibits the growth of hepatoma cells by blocking the cell cycle or inducing apoptosis, we examined the cell cycle distribution and apoptosis of PLC/PRF/5 cells after DHW treatment. Among the cell lines we examined, PLC/PRF/5 cell is the most sensitive to DHW treatment (IC$_{50}$ value: 43.29 ± 1.34 $\mu$g/ml for 72 hrs treatment, Table 1). PLC/PRF/5 cells were cultured for 24 hrs, then treated with 40 $\mu$g/ml DHW extract for 48 hrs, and analyzed by flow cytometry. We found that the number of cells arrested in the G0/G1 phase increased by 20%, while the proportion of cells in the S phase decreased sharply by 16% (Fig 1D). As shown in Fig 1E, the apoptosis of PLC/PRF/5 cells increased only slightly from 2.5 ± 0.4% to 3.8 ± 0.7% after 48 hrs of treatment with 40 $\mu$g/ml DHW extract. This result indicates that DHW does not inhibit PLC/PRF/5 cells by inducing apoptosis. With EDUrd incorporation assay, we further confirmed that DHW extract significantly inhibited DNA synthesis in PLC/PRF/5 cells (Fig 1F). The result in Fig 1 demonstrates that DHW extract inhibits the proliferation of hepatoma cells.

### Berberine and Costunolide, two key active components in DHW, synergistically inhibit hepatoma cell proliferation

DHW is a traditional herbal medicine consisting of Coptidis Rhizoma and Aucklandia lappa Decne, with a ratio of 1:1 (w/w). It is well known that Berberine and Costunolide are the main

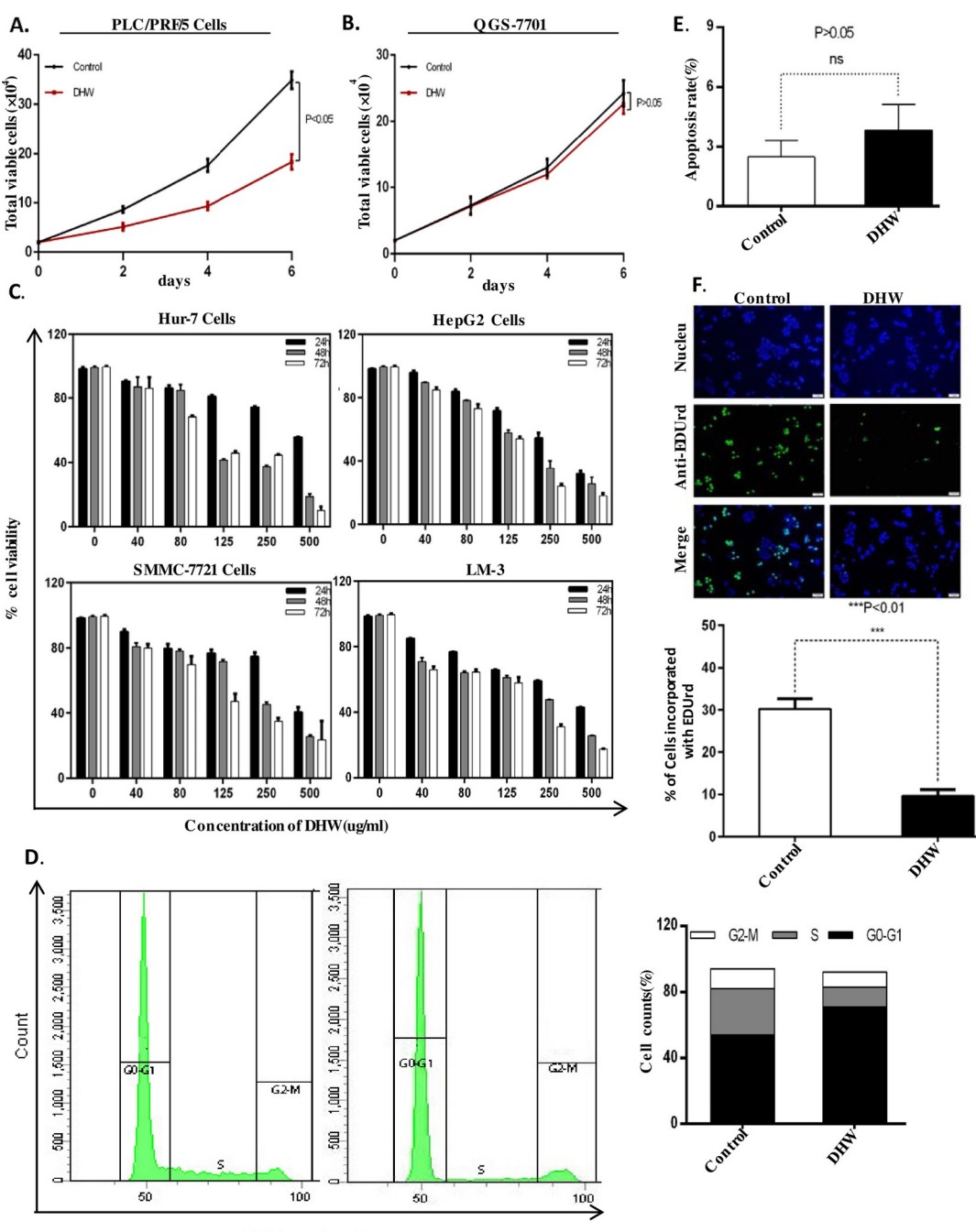

**Fig 1.** PLC/PRF/5 (A), and QSG-7701 (B), cells ($2\times10^4$) were seeded in 12-well plates and cultured in DMEM treated with DHW (40 $\mu$g/ml). The number of cells was counted at the indicated times. C, Cell viability determined by MTT assay. Liver cancer cells were treated with different concentrations of DHW extract for 24, 48, and 72 hrs, and cell viability was determined by MTT. D, DHW extract induces G0/G1 arrest of PLC/PRF/5 cell line. Cells were treated without or with 40 $\mu$g/ml DHW extract for 48 hrs and then the cell cycle population was measured by flow cytometry. E, Apoptosis in PLC/PRF/5 cells determined by flow cytometry. Cells were treated without or with 40 $\mu$g/ml DHW extract for 48 hrs, and then apoptosis was measured by flow cytometric analysis of cells stained with Houchest3324 and PI. The percentages of Houchest3324 or/and PI-positive cells are indicated. F, PLC/PRF/5 Cells were treated with DHW extract for 48 hrs before they were assayed for EdUrd incorporation. Data are presented as the mean ±SD of values from triplicate experiments.

Table 1. IC$_{50}$ values of DHW extract on different cell lines.

| Cell line | IC$_{50}$ values (µg/ml) | | |
| --- | --- | --- | --- |
| | 24h | 48h | 72h |
| PLC/PRF/5 | 213.10±1.07 | 68.95±1.42 | 43.29±1.34 |
| HepG2 | 376.30±0.78 | 178.90±0.98 | 119.90±0.84 |
| LM3 | 438.60±0.89 | 150.90±1.06 | 128.80±1.11 |
| Hur7 | 338.50±1.06. | 212.80±1.05 | 171.50±0.95 |
| SMMC7721 | 341.20±1.00 | 220.70±1.25 | 147.14±1.17 |

bioactive components of Coptidis Rhizoma and Aucklandia lappa Decne [29]. Therefore, we first quantified the content of Berberine (5.6%) and Costunolide (1.2%) in DHW extract by HPLC (Fig 2A and 2B). Then, pure Berberine (2.5 µg/ml) and Costunolide (0.5 µg/ml) were mixed according the abundance determined above to form a Berberine-Costunolide mixture (named as BC mix here after). PLC/PRF/5 cells were incubated with Berberine (2.5 µg/ml), Costunolide (0.5 µg/ml), this BC mix, or 40 µg/ml DHW extract for 48 hrs. EDUrd incorporation assay was then performed. As shown in Fig 2C, both Berberine and Costunolide inhibited the proliferation of PLC/PRF/5 cells. The BC mix and DHW extract had similar inhibitory effects on PLC/PRF/5 cells (Fig 2C). The inhibitory effect of BC mixture or DHW extract on PLC/ PRF/5 cells was better than that of Berberine or Costunolide alone (Fig 2C). The results indicate that Berberine and Costunolide are indeed anti-cancer active ingredients of DHW, and the combination of Berberine and Costunolide is sufficient to mimic the inhibition of DHW on PLC/PRF/5 cells.

In these experiments (Fig 2C), we noted that the inhibitory effect of BC mix on PLC/PRF/5 cell proliferation was greater than the sum of the inhibitory effects of Berberine and Costunolide. To evaluate the combined effects of Berberine and Costunolide on PLC/PRF/5 cells, we treated PLC/PRF/5 cells with different concentrations of BC mix. As shown in Fig 2D, the treatment with BC mix was more effective in inhibiting the proliferation of PLC/PRF/5 cells than the sum of the inhibitory effects of using Berberine or Costunolide alone. The results showed that the combination of Berberine and Costunolide has a synergistic effect on inhibiting the proliferation of hepatoma cells. Subsequently, we used the Calcusyn software program for median analysis. The median effect analysis was then performed using the Calcusyn software program. The effect of constant ratio of Berberine and Costunolide on the growth of the PLC/PRF/5 cells was calculated. The data obtained in this study were analyzed by the Chou-Talalay method, and the synergistic effects were analyzed by the combination index (CI). CI $<1$, $= 1$, and $>1$ represent synergistic, additive, and antagonistic effects, respectively [30]. We found that four of the five BC mix concentrations used in this experiment had a combined index of less than 1 (CI$<1$) (Fig 2E), indicating that the Berberine and Costunolide have synergistic anti-tumor effects (Table 2). This synergistic effect brings the extra benefit of a combination of Coptidis Rhizoma and Aucklandia lappa Decne, which supports the value of DHW formula.

## DHW inhibits the proliferation of hepatoma cells by regulating EGF signaling pathway

We set out to explore the potential mechanism by which DHW inhibits the proliferation of liver cancer cells. EGF signaling is a core signaling pathway that regulates cell proliferation. Overexpression of EGFR and its abnormal activity are closely related to the occurrence and development of various malignant tumors including liver cancer. Therefore, we first

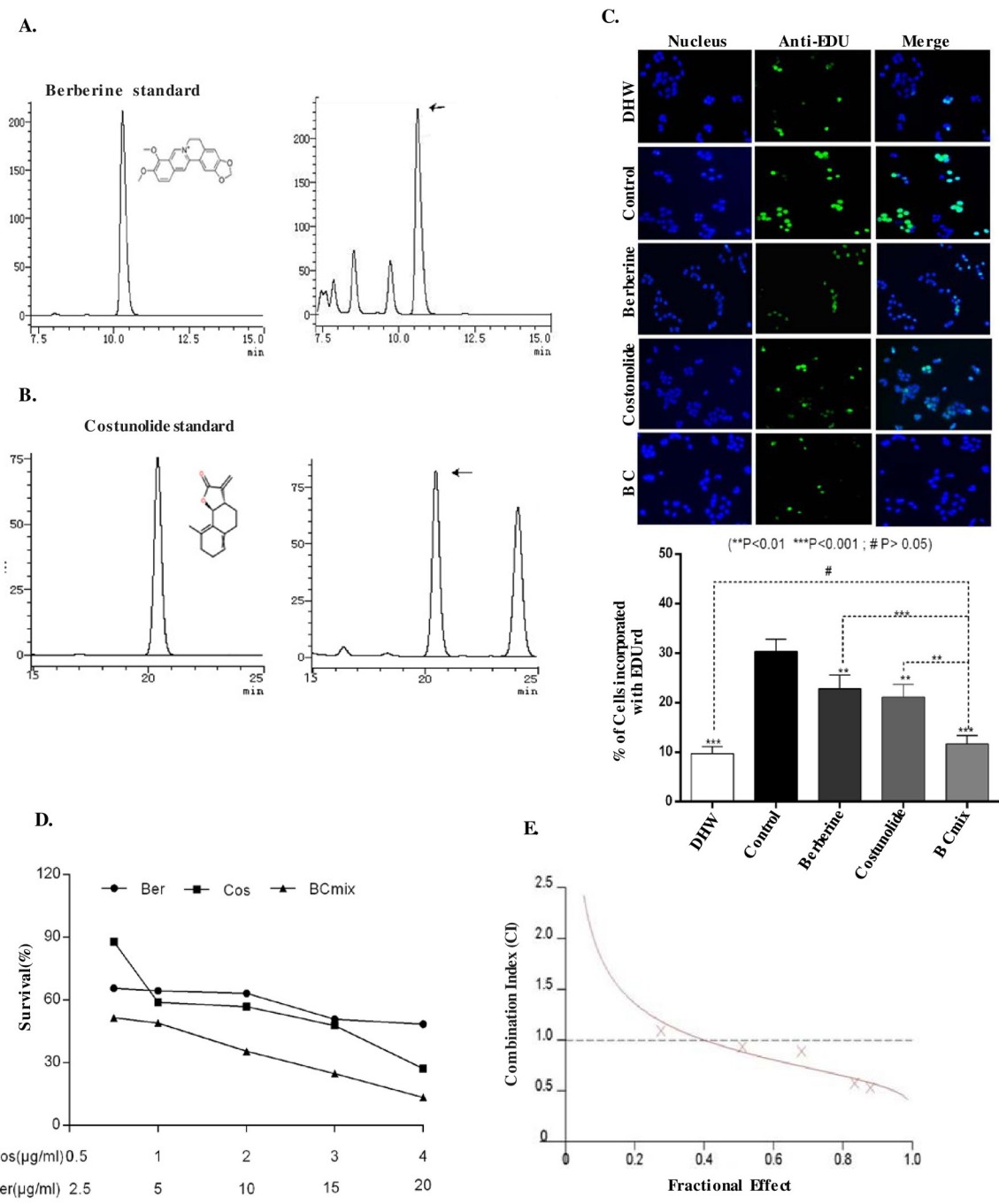

**Fig 2.** A, Panel 1 is the chromatogram and structure of Berberine standard; Panel 2 is the chromatogram of DHW samples; High performance liquid chromatograms. B, Panel 1 is the structure and chromatogram of Costunolide standard, Panel 2 is the chromatogram of DHW samples in different analysis. Arrows indicate the peaks representing Berberine or Costunolide; C, EdUrd incorporation assays were performed on PLC/PRF/5 cells treated with either DMSO (control), Berberine (2.5 $\mu$g/ml) and/or Costunolide (0.5 $\mu$g/ml), DHW extract (40 $\mu$g/ml). The percentage of cells incorporating EDUrd after 48 hrs was shown. D, PLC/PRF/5 cells were treated with Berberine (Ber) and Costunolide(Cos) or both (BC mix) in a fixed ratio (5:1) for 48 hrs. Cell viability was measured by MTT assay. E, CI was calculated by isobologram analysis using the Chou-Talalay method. CI = 1, additive effect; CI <1, synergistic effect; CI, >1, antagonistic effect. Data represented are from three independent experiments.

**Table 2. Synergistic effects of Berberine with Costunolide on growth of PLC/PRF/ 5 cells.**

| Dose (μg/mL) | Fraction of Cells Affected (FA) | | | CI |
|:---:|:---:|:---:|:---:|:---:|
| | **Berberine** | **Costunolide** | **BC mix** | |
| 2.5 | 0.168 | 0.153 | 0.276 | 1.095 |
| 5.0 | 0.394 | 0.436 | 0.509 | 0.935 |
| 10 | 0.405 | 0.449 | 0.681 | 0.895 |
| 15 | 0.688 | 0.520 | 0.834 | 0.578 |
| 20 | 0.778 | 0.531 | 0.880 | 0.533 |

Abbreviations are as follows: FA, Fraction of cells with growth affectedin response to Berberine and /or Costunolide treated vs. uncreated cells; CI, combination Index.
CI < 1.0 indicates synergism.

determined the expression and activation of EGFR in hepatoma cells such as PLC/PRF/5, HCC-LM3, HepG2, Hur7, SMMC-7721 cells. QSG-7701 human hepatocyte was used as controls. As shown in Fig 3A, both EGFR expression and phosphorylation were significantly increased in hepatoma cells compared to normal hepatocytes (Fig 3A, 1st panel from top, lanes 1–5, compared to lane 6). We further confirmed that EGF signaling is critical for the proliferation of liver cancer cells by EGFR inhibitor SC0186. PLC/PRF/5, LM3, HepG2, Hur7, SMMC-7721 cells were cultured and then treated with 10mM SC0186 for 24 hrs. The proliferation of these cells was inhibited by SC0186, and EGFR phosphorylation of these cells was also blocked (Fig 3B, 1st panel from top, lane 1–5).

Since EGF signaling is critical for the proliferation of hepatoma cells, we then further explored whether DHW inhibits hepatoma cell proliferation by regulating EGF signaling. PLC/PRF/5 cells were cultured and incubated with DHW extract, Berberine, Costunolide, or BC mix for 48 hrs. As shown in Fig 3, the expression and phosphorylation of EGFR were significantly reduced after treatment with DHW extract or BC mix (Fig 3C, 1st Panel from top, lane 1 and 5 compared to lane 2; 2nd Panel from top, lane 1 and 5 compared to lane 2). Examination on the activities of the downstream kinases of EGF signaling revealed that the Protein Kinase B (AKT) phosphorylation was remarkably inhibited, whereas Extracellular signal-regulated kinase (ERK) phosphorylation was up-regulated (Fig 3C, 1st panel from top, lane 6 and 10 compared to lane 7; Fig 3C, 1st panel from top, lane 11 and 15 compared to lane 12). We further confirmed this result with the EGFR inhibitor SC0186. Both DHW extract and the EGFR inhibitor SC0186 significantly inhibited the growth of hepatoma cells (Fig 3D). However, DHW extract alone and the combination of DHW extract and SC0186 have the similar inhibition on PLC/PRF/5 cell proliferation (Fig 3D). We believe that this is because DHW inhibits cell proliferation primarily through EGF signaling, and the use of the EGFR inhibitor SC0186 in combination with DHW does not provide additional benefits. Therefore, this result indicates that DHW inhibits hepatoma cell proliferation by modulating EGF signaling.

## Berberine and Costunolide promote the endocytosis and degradation of EGFR respectively

EGF signaling is the core of cell proliferation and is always tightly controlled. Upon activated by extracellular signals, phosphorylated EGFR is endocytosed through either clathrin-dependent or clathrin-independent pathways [26]. When EGFR is endocytosed, it is then ubiquitinated and directed to lysosomes for degradation. There are also some EGFR that will be recycled back to the cell surface to participate in the new activation process. Endocytosis and degradation of EGFR shut down the activated EGF signaling, allowing for tight regulation of the EGF signaling pathway. In addition to, for example, mutation-induced

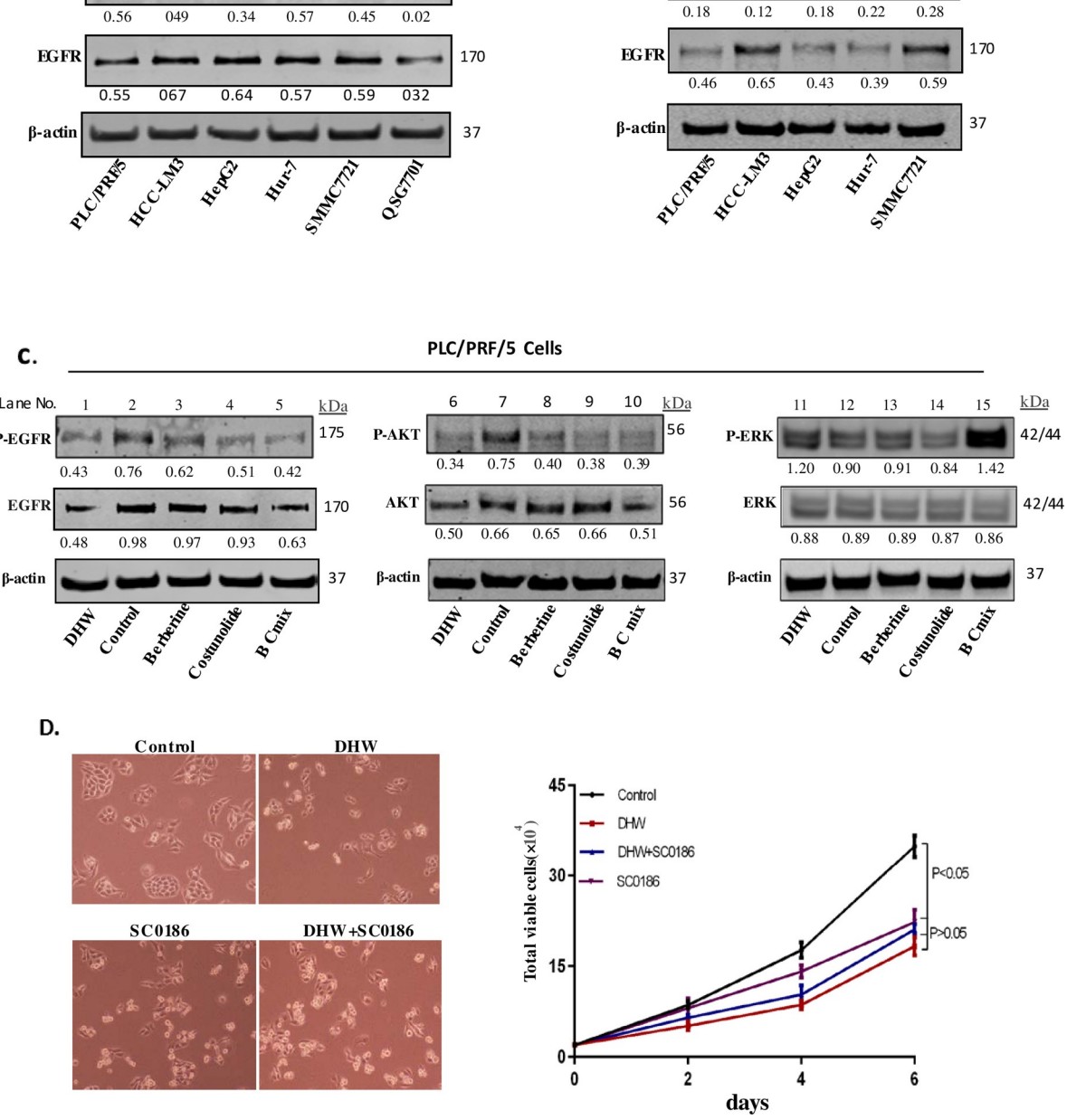

**Fig 3.** A, the expression and activation of EGFR in hepatoma cells such as PLC/PRF/5, HCC-LM3, HepG2, Hur7 and SMMC-7721 cells compared to normal hepatocytes; B, The EGFR and P-EGFR levels in HCC cells were examined by Western blot after treatment treated with 10mM SC0186 for 24 hrs; C, PLC/PRF/5 cells were treated with DHW extract, Berberine, Costunolide or BC mix at concentrations of 40 μg/ml, 2.5 μg/ml and 0.5 μg/ml for 48 hrs, and then the expression levels of the relevant factors (EGFR, P-EGFR, Akt, P-Akt, ERK, P-ERK) were determined by Western blot; D, Effect of DHW extract, he EGFR inhibitor SC0186 and the combination of DHW extract and SC0186 on cellular growth and morphological changes of PLC/PRF/5 cells.

hyperphosphorylation of EGFR, the abnormalities in EGFR signaling are more due to the disruption of the balance of EGFR endocytosis and degradation.

In this study, we found that DHW inhibits hepatoma cell proliferation by blocking EGF signaling (Fig 3C). Therefore, we first examined whether Berberine and Costunolide inhibit EGF signaling by directly blocking EGFR phosphorylation. PLC/PFR/5 cells were serum-starved and incubated with Berberine, Costunolide, BC mix, or DHW extract for 24 hrs, and then stimulated by EGF (10 ng/ml) for 30 mins. As shown in Fig 3C, although both Berberine and Costunolide reduced EGFR phosphorylation in PLC/PFR/5 cells cultured in full serum (Fig 3C, 1st panel from top, lane 3 and 4 compared to lane 2), but neither Berberine nor Costunolide blocked EGFR phosphorylation stimulated by EGF (Fig 4A, 1st panel from top, lane 8 and 9 compared to lane 7). DHW extract or BC mix also failed to prevent EGF-stimulated EGFR phosphorylation (Fig 4A, 1st panel from top, lane 6 and 10 compared to lane 7), while both inhibited EGFR phosphorylation in cells cultured in full serum (Fig 3C, 1st panel from top, lane 1and 5 compared to lane 2). We also noticed that DHW extract treatment significantly reduced EGFR expression in cells cultured in full serum (Fig 3C, 2nd Panel from top, lane1 compared to lane 2). All of these results indicate that DHW or its major active ingredient, Berberine or Costunolide, does not directly target EGFR phosphorylation, but indirectly reduces EGF signaling over a longer period of time. Therefore, we suspect that Berberine or Costunolide might target the endocytosis and degradation processes of EGFR.

We set out to examine the effects of Berberine or Costunolide on the endocytosis and ubiquitination of EGFR. We used immunofluorescence to detect the presence of EGFR on the surface of PLC/PRF/5 cells. As shown in Fig 4B, Berberine, Costunolide, or BC mix attenuated the presence of EGFR on the surface of PLC/PRF/5 cells (Fig 4B, picture from left, picture 2–4 compared to picture 1). We then isolated cell surface EGFR using EZ-link Sulfo–NHS–SS–biotin. Costunolide and BC mix significantly reduced the amount of EGFR on the cell surface (Fig 4C, 1st panel from top, lane6 and 8 compared to lane 5). The reduced cell surface EGFR may be due to a decrease of EGFR precursor transport resulting in reduced amount of cytoplasmic protein to the cell surface, or the accelerated endocytosis of EGFR from the cell surface. Therefore, we examined the effect of Berberine or Costunolide on the EGFR internalization after EGF stimulation. We first labeled the cell surface proteins with Sulfo–NHS–SS–biotin and then incubated these PLC/PRF/5 cells with Berberine, Costunolide, or BC mix as indicated. The biotin label of the cell surface proteins were then stripped with MesNa. Therefore, the signals we detected represent internalized cell surface proteins. As shown in Fig 4, after Berberine treatment, the internalization of cell surface EGFR was enhanced, while Costunolide had no significant effect on the internalization of EGFR (Fig 4C, 1st panel from top, lane10 and lane11 compared to lane 9). This result indicates that Berberine promotes the internalization of EGFR. Subsequently, we evaluated EGFR ubiquitination in PLC/PRF/5 cells exposed to Berberine, Costunolide, BC mix via immunoprecipitation with anti-EGFR antibody followed by immunoblotting with an anti-ubiquitin antibody. We found that Costunolide and BC increased EGFR ubiquitination, while Berberine has a limited effect on EGFR ubiquitination (Fig 4D, 1st panel from top, lane 2, 3 and 4 compared to lane 1). These results demonstrated that Berberine and Costunolide target different processes of EGFR degradation. Berberine promotes the internalization of EGFR while Costunolide can increase EGFR ubiquitination. This also explains why the combination of Berberine and Costunolide has a synergistic effect on EGF signaling, thereby providing DHW with an additional advantage of inhibiting cancer cell proliferation and treating cancer.

## Discussion

DHW is a traditional herbal medicine used by Mongolians for many years. Although the underlying mechanisms are unclear, the drug does benefit cancer patients. For many

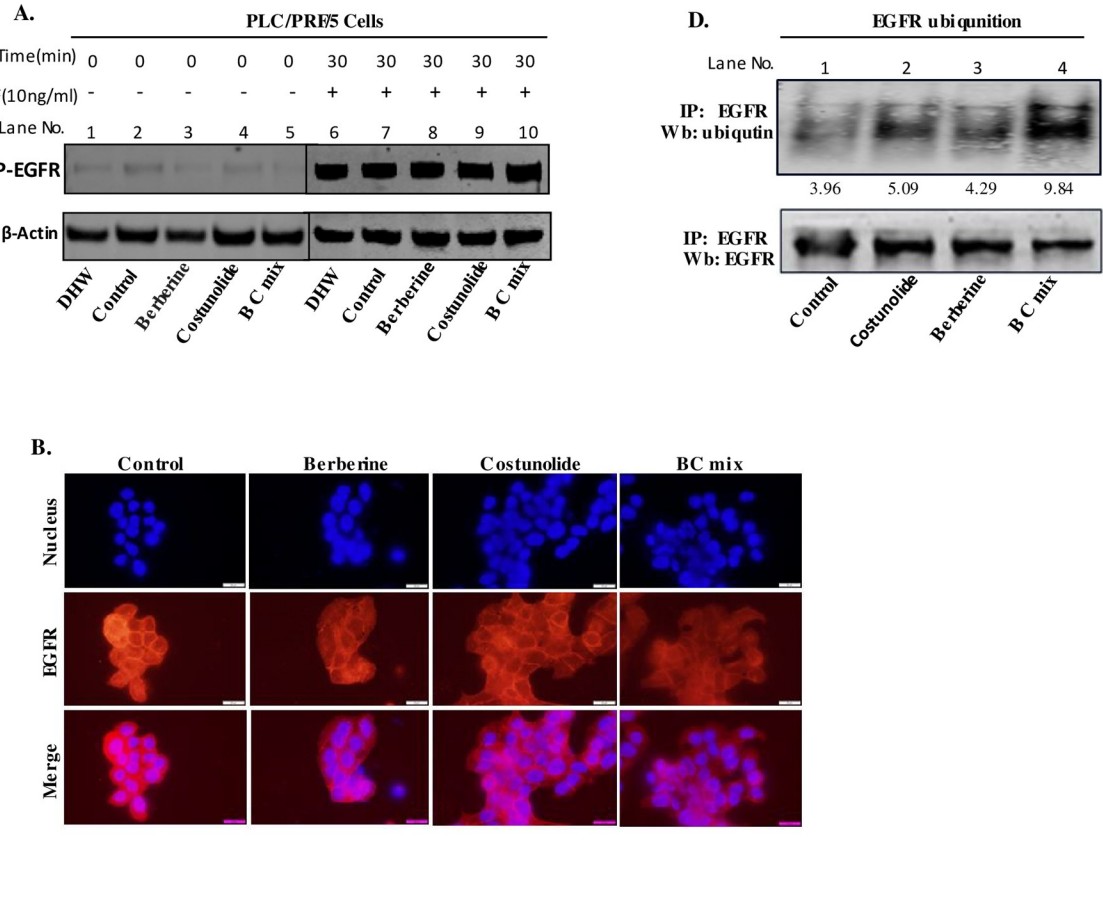

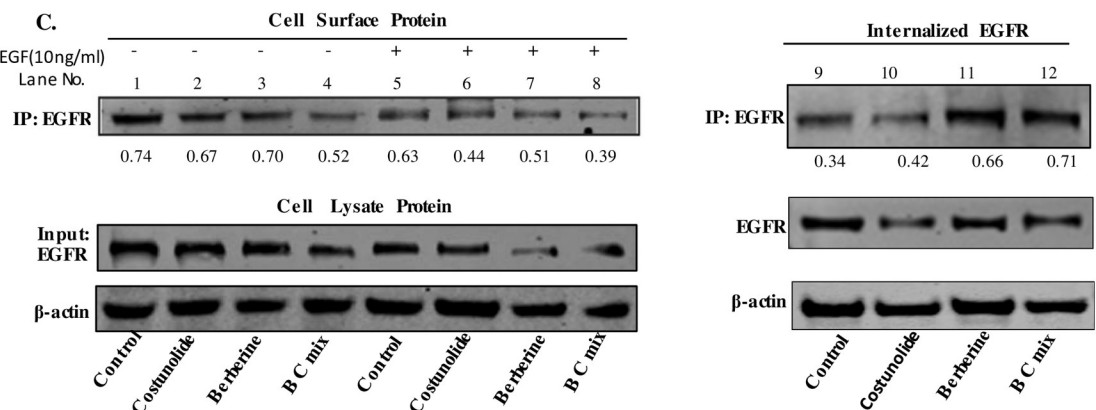

**Fig 4.** A, PLC/PFR/5 cells were serum-starved and incubated without (control) or with Berberine, Costunolide and BC mix for 24 hrs, and then stimulated by EGF (10ng/ml) for 30 mins. The expression level of P-EGFR in PLC/PRF/5 cells was examined by Western blot. B, EGFR on the surface of PLC/PRF/5 cells were examined by immunofluorescence after treated with Berberine, Costunolide and BC mix for 24 hrs, and then stimulated by EGF (10ng/ml) for 30 mins; C, PLC/PFR/5 cells were serum-starved and incubated without (control) or with Berberine, Costunolide, and BC mix for 24 hrs, and then stimulated by EGF (10ng/ml) for 30 mins. Proteins on cell surface were labeled with Sulfo-NHS-SS-Biotin and isolated by streptavidin and then analyzed by immunoblot detected EGFR (lane5-8); The internalized cell surface proteins were isolated using Sulfo-NHS-SS-Biotin and MesNa and analyzed by immunoblot detecting EGFR (top panel) (lane9-12). D, PLC/PRF/5 cells were incubated without (control) or with Berberine, Costunolide, and BC mix for the indicated time periods. Ubiquitinated EGFR was examined via EGFR immunoprecipitation followed by immunoblotting with an ubiquitin antibody. The bands were quantified. The level of ubiquitinated EGFR was normalized to the immunoprecipitated EGFR. The relative density of the control.

traditional herbal medicines, the situation is similar: there are no well-controlled scientific experiments to verify the validity of these drugs, and there are no clues about their underlying mechanisms and side effects, which greatly limits clinical applications and further optimization of these drugs. In this study, we first demonstrated that DHW inhibits the proliferation of hepatoma cells by modulating EGF signaling. Berberine and Costunolide are the main active ingredients in DHW. Berberine promotes endocytosis of activated EGFR, while Costunolide increases ubiquitination of EGFR and reduces the distribution of EGFR on the cell membrane, thereby inhibiting the activation of EGF signaling. Berberine and Costunolide target different processes of EGFR degradation, resulting in a significant synergistic effect in inhibiting cell proliferation. This is the benefit of a combination of Berberine and Costunolide in the DHW formula. In addition, neither Berberine nor Costunolide directly target EGFR phosphorylation. This allows DHW to be used as a supplementary medicine to tyrosine kinase inhibitors in cancer therapy.

EGF signaling is a key signaling pathway that regulates cell proliferation. Extracellular signal molecules stimulate EGFR phosphorylation and then trigger this signal cascade. Phosphorylation of EGFR also initiates the internalization of EGFR, thereby shutting down the EGF signaling. Phosphorylated EGFR will then be ubiquitinated and engulfed by primary endocytosis, transforming from a membrane distribution to an intracytoplasmic vesicle distribution. Primary endocytic vesicles grow into secondary vesicles and bind to lysosomes, and the ubiquitin-protease system can be activated to degrade EGFR [24,31]. We found that both Berberine and Costunolide target and facilitate the EGFR internalization and degradation system. The synergistic effect of the combination of Berberine and Costunolide suggests that they target different EGFR internalization and degradation processes. We further demonstrate that Berberine promotes endocytosis of activated EGFR, while Costunolide increases ubiquitination of EGFR. However, the exactly targets of Berberine and Costunolide remain unclear. To identify these targets and underlying mechanisms, further experiments are necessary in the future.

In this study, we focused on the effects of Berberine and Costunolide on EGF signaling. As natural compounds, Berberine and Costunolide may target other signaling cascades. Previous studies have shown that Berberine prevents secretion of VEGF from HCC and down-regulating VEGF mRNA expression [32]. Costunolide also inhibits proliferation and survival of colorectal cancer cells via inhibiting Wnt/β-Catenin signaling pathway [33]. Although we cannot rule out the possibility of Berberine and Costunolide targeting other signaling pathways, we did demonstrate in this study that in certain tumor cells, like PLC/PRF/5, HT29, and HCC-LM3, Berberine and Costunolide inhibit tumor cell proliferation by modulating EGF signaling.

DHW is a traditional herbal medicine. In this study, we determined that Berberine and Costunolide are the main active components in DHW. This does not exclude the presence of other active ingredients in the DHW. In fact, Coptidis Rhizoma is composed of various alkaloids, including Berberine, coptisine, palmatine, and jatrorrhizine. Aucklandia lappa Decne root extract contains resinoids, essential oil, alkaloid, inulin, a fixed oil, and other minor ingredients such as tannins and sugars [12]. In this study, we found that BC mix and DHW extract have similar inhibitory effects on PLC/PRF/5 cell proliferation (Fig 2C). They inhibited PLC/PRF/5 cells better than either Berberine or Costunolide alone (Fig 2D). Therefore, at least for certain liver cancer cell such as PLC/PRF/5 cells, Berberine and Costunolide are indeed the main anti-cancer active ingredients of DHW.

As summarized in Fig 5, our study indicates that Berberine and Costunolide are key active ingredients for inhibiting the growth of hepatoma cell by DHW extract *in vitro*. Berberine and Costunolide target different steps of EGFR internalization and degradation, and mutually

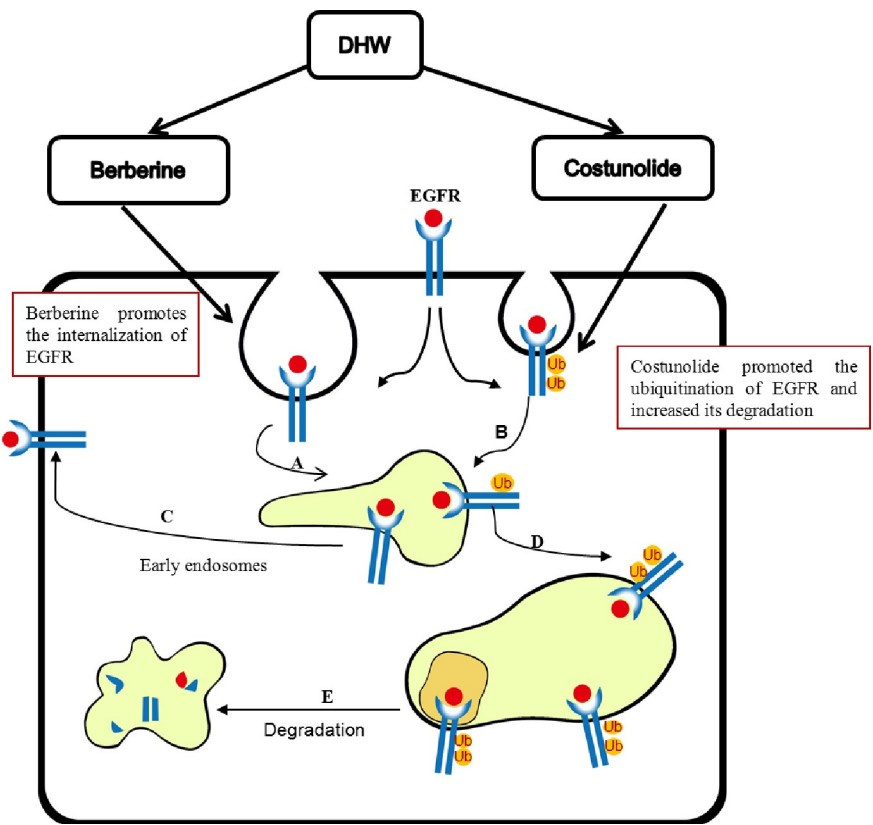

**Fig 5. A model for the function of DHW in EGFR homeostasis.** Berberine and Costunolide are the main active ingredients in DHW. Berberine promotes endocytosis of activated EGFR, while as Costunolide increases ubiquitination of EGFR and reduces EGFR recycling to cell membrane distribution, thereby inhibiting EGF signaling. Part of internalized EGFR goes into early endosomes (A). Following their internalization into early endosomes, nonubiquitinated receptors are recycled back to the plasma membrane (C). Part of internalized EGFR is ubiquitinated (B). Ubiquitinated EGFRs proceed toward the multivesicular bodies (D). Ubiquitinated cargo pour into lysosomes for degradation (E).

enhance the inhibition on EGF signaling, resulting in a synergistic anti-cancer effect (Fig 5). These results suggest that the combination of Berberine and Costunolide is a potential anti-tumor drug for the treatment of liver cancer, and also provides a new understanding of the mechanism of action of traditional Mongolian medicines.

## Supporting information

**S1 File.**
(PDF)

## Acknowledgments

We acknowledge members of the Hongyang laboratory for critically reading the manuscript.

## Author Contributions

**Conceptualization:** Ba Genna, B. Burenbatu, Qiyu Feng, Hongyang Wang.

**Data curation:** Ha Si.

**Formal analysis:** Ha Si, Qiyu Feng.

**Funding acquisition:** Ha Si, Ba Genna, Qiyu Feng, Hongyang Wang.

**Investigation:** Ha Si, Xiangjin Zhuang.

**Methodology:** Qiyu Feng.

**Project administration:** Qiyu Feng.

**Resources:** Ha Si, B. Burenbatu, Qiyu Feng, Hongyang Wang.

**Software:** Ha Si, Jing Wang.

**Supervision:** Ba Genna, Qiyu Feng, Hongyang Wang.

**Validation:** Xiangjin Zhuang, Jing Wang.

**Writing – original draft:** Ha Si.

**Writing – review & editing:** Ha Si, Qiyu Feng.

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
