## [Decision Letter · Decision Letter 0]

27 Jan 2020

PONE-D-19-35686

DaHuangWan targets EGF signaling to inhibit the proliferation of hepatoma cells

PLOS ONE

Dear Dr. Feng,

Thank you for submitting your manuscript to PLOS ONE. After careful consideration, we feel that it has merit but does not fully meet PLOS ONE’s publication criteria as it currently stands. Therefore, we invite you to submit a revised version of the manuscript that addresses the points raised during the review process.

Both reviewers have raised issues which should be addressed if the authors intend to submit a revised manuscript.  With respect to Reviewer 2, it is true that there are many other possible growth factor-dependent pathways that could be studied.  It is not unreasonable to have chosen EGF.  The other questions raised are more at issue.

We would appreciate receiving your revised manuscript by Mar 12 2020 11:59PM. To enhance the reproducibility of your results, we recommend that if applicable you deposit your laboratory protocols in protocols.io, where a protocol can be assigned its own identifier (DOI) such that it can be cited independently in the future. For instructions see: http://journals.plos.org/plosone/s/submission-guidelines#loc-laboratory-protocols

We look forward to receiving your revised manuscript.

Kind regards,

Salvatore V Pizzo

Academic Editor

PLOS ONE

5. Please include your tables as part of your main manuscript and remove the individual files. Please note that supplementary tables (should remain/ be uploaded) as separate "supporting information" files

Reviewers' comments:

Reviewer's Responses to Questions

**Comments to the Author**

1. Is the manuscript technically sound, and do the data support the conclusions?

Reviewer #1: Yes

Reviewer #2: Yes

2. Has the statistical analysis been performed appropriately and rigorously? 

Reviewer #1: Yes

Reviewer #2: N/A

3. Have the authors made all data underlying the findings in their manuscript fully available?

Reviewer #1: Yes

Reviewer #2: Yes

4. Is the manuscript presented in an intelligible fashion and written in standard English?

Reviewer #1: Yes

Reviewer #2: Yes

5. Review Comments to the Author

Reviewer #1: The manuscript named “DaHuangWan targets EGF signaling to inhibit the proliferation of hepatoma cells” identifies the main active ingredients in DaHuangWan (DHW) and reveals the underlying mechanisms of treating liver cancer. It also provides basis for reveals the mechanisms of herb pair. Overall the manuscript is a generally well presented paper with a good design. However, I think there are still some areas that need to be improved to achieve the quality of publication in this journal. I would recommend the paper for publication with minor revisions as addressed in the comments section bellow.

Question 1: How to determine the concentration of DHW extract is 40 μg/mL and the treat time is 48 h? When the patient is treated with DHW normally, can the effective physiological concentration in the body reach 40 μg/mL?

Question 2: Page 6, Line 116, whether the concentration range of DHW extract (40~500 μg/mL) is non-toxic to QSG-7701 human hepatocytes?

Question 3: Page 6, Line 117-118, “The IC50 values of these liver cancer cells was 118 calculated.”, please add the Table1 for better illustration.

Question 4: Figure 2C, there are two DHW, please confirm.

Question 5: Figure 2D, the abscissa position is offset, please correct.

Question 6: Figure 4B, why is there no “merge” picture for better illustration?

Question 7: Table 2, “Dose” is berberine dose, but the title is “Synergistic effects of Berberine with Costunolide on growth of PLC/PRF/ 5 cells”, please make it clear.

Question 8: Page 7, Line 143, which is the final concentrations of berberine and costunolide respectively in the BC mix?

Question 9: Page 8, Line 165, where is the Fig.2E, please confirm.

Question 10: Please check all unit writing forms, such as “40 μg/mL” not “40μg/mL”, “10 ng/ml” not “10ng/ml”, “30 mins” not “30mins” and so on.

Question 11: Please unify the time form, “hrs”, “hours”, “h”.

Question 12: Please confirm whether the berberine and costunolide initials need to be capitalized, the text is inconsistent.

Question 13: Page 16, Line 349, please change “of 40 g/ml After” to “of 40 g/ml. After”.

Question 14: Page 19, Line 418, “DMEM supplemented with DHW (40μg/ml).” , please unify the font form.

Question 15: Page 20, Line 458, please change “(control)or” to “(control) or”.

Question 16: Please check some latest published articles and modify the format of the references according the guidelines. Such as “Page 21, Line 470, Please add the issue number.”

Question 17: Page 21, Line 492, Please check the page number.

Question 18: Page 22, Line 523, Please add the page number.

Question 19: Page 22, Line 536, Please check the page number.

Question 20: I recommend that the author add a model diagram of the mechanism for readers to better understand.

Reviewer #2: The study investigates a Mongolian herbal medicine DaHuangWan in inhibiting hepatoma cell growth through inhibiting EGF signalling. However the following questions should be addressed before acceptance.

1. There are so many signalling pathways related to cancer cell proliferation. Why did the authors target on EGFR signalling?

2. Did the authors consider studying the anti-proliferative effect of other components in DHW? And how the authors exclude the possibility of combined toxicity of costunolide and berberine in the B+C mixture rather than synergistic effect?

3. To confirm the DHW-regulated EGF signalling, it is more reasonable to rescue the EGFR expression rather than blocking it by inhibitor.

4. It is interesting to show that berberine promotes endocytosis while costunolide promotes EGFR ubiquitination. But how did the authors explain for Fig 4D that berberine seems increase the ubiquitin level of EGFR.

6. PLOS authors have the option to publish the peer review history of their article (what does this mean?). If published, this will include your full peer review and any attached files.

Reviewer #1: No

Reviewer #2: No

---

## [Author Response · Author response to Decision Letter 0]

8 Mar 2020

Reviewer #1: The manuscript named “DaHuangWan targets EGF signaling to inhibit the proliferation of hepatoma cells” identifies the main active ingredients in DaHuangWan (DHW) and reveals the underlying mechanisms of treating liver cancer. It also provides basis for reveals the mechanisms of herb pair. Overall the manuscript is a generally well presented paper with a good design. However, I think there are still some areas that need to be improved to achieve the quality of publication in this journal. I would recommend the paper for publication with minor revisions as addressed in the comments section bellow.

Question 1: How to determine the concentration of DHW extract is 40 μg/mL and the treat time is 48 h? When the patient is treated with DHW normally, can the effective physiological concentration in the body reach 40 μg/mL?

In this study, we first examined the effects of DHW extract (10 ~ 500 ug/ml) on the growth of various liver cancer cells. We found that 40 μg/ml DHW extract significantly inhibited the growth of PLC/PRF/5 cells while being non-toxic to QSG-7701 human hepatocytes (Figures 1A and 1B). Therefore, we consider 40 μg/ml DHW extract to be appropriate because it clearly demonstrates its effect on the proliferation of liver cancer cells and excludes the possibility of DHW extract toxicity. In addition, 40μg/ml DHW extract has around 2.5 ug/ml (~ 8 uM) Berberine and 0.5 ug/ml Costunolide. 10 – 30 uM berberine is commonly used for cell signal transduction studies through cell culture (e.g. Wang J et al. Oncotarget. November 15, 2016; 7 (46): 76076-76086).

The effective physiological concentration of DHW in the body has not been determined. However, physiological concentrations of Berberine have been reported. The 40 μg/ml DHW extract has about 2.5 ug/ml (~8 uM) Berberine and 0.5 ug/ml Costunolide. Chen J et al. reported that after oral administration of Coptidis Rhizoma extract (0.15 g / 100 g body weight, containing 3.1% small base), Berberine is absorbed at a higher absorption rate, with a Tmax value of 58.21 ± 9.9 minutes and a Cmax value of 2.9 ± 1.1 μg/ml (Chen J et al. Bull. Korean Chem. Soc. 2013,5(34):1559-1562). Therefore, we believe that the effective physiological concentration of DHW extract in the body might reach 40 μg/ml.

Question 2: Page 6, Line 116, whether the concentration range of DHW extract (40~500 μg/mL) is non-toxic to QSG-7701 human hepatocytes?

Yes, we examined the effects of DHW extract (concentration range of 10 ~ 500 ug/ml) on the growth of various liver cancer cells and QSG-7701 human hepatocytes. DHW extract inhibited the growth of SMMC-7721, HepG2, Hur7, and LM3 cells in a dose-and time-dependent manner, and these cells had different sensitivities against DHW extract (Figure 1C). Moreover, DHW extract (10~500μg/mL) is non-toxic to QSG-7701 human hepatocytes.

Question 3: Page 6, Line 117-118, “The IC50 values of these liver cancer cells was 118 calculated.”, please add the Table1 for better illustration.

We have added the Table 1 as requested by the reviewer. Please see Page 8, Line 232.

Question 4: Figure 2C, there are two DHW, please confirm.

We have fixed it as requested by the reviewer. Please see Figure 2C.

Question 5: Figure 2D, the abscissa position is offset, please correct.

We have fixed it as requested by the reviewer. Please see Figure 2D.

Question 6: Figure 4B, why is there no “merge” picture for better illustration?

The merged pictures have been added as requested by the reviewer. 

Question 7: Table 2, “Dose” is berberine dose, but the title is “Synergistic effects of Berberine with Costunolide on growth of PLC/PRF/ 5 cells”, please make it clear.

Synergies were analyzed by CalcuSyn software. It is pointed out in the CalcuSyn software usage guide: Enter the two (or more) drugs that form the combination/mixture using the approach for entering single drugs (e.g. Adhip P. N. et al. Nutrition and Cancer, 61(4), 544–553)

Question 8: Page 7, Line 143, which is the final concentrations of berberine and costunolide respectively in the BC mix?

The final concentrations of Berberine and Costunolide in BC mix are 2.5 𝜇g/ml and 0.5 𝜇g/ml respectively (Line 240-242).

Question 9: Page 8, Line 165, where is the Fig.2E, please confirm.

We have confirmed it (Line 263 now) as requested. 

Question 10: Please check all unit writing forms, such as “40 μg/mL” not “40μg/mL”, “10 ng/ml” not “10ng/ml”, “30 mins” not “30mins” and so on. 

We have checked and revised all unit writing forms as requested.

Question 11: Please unify the time form, “hrs”, “hours”, “h”.

The time form has been unified to “hrs”. 

Question 12: Please confirm whether the berberine and costunolide initials need to be capitalized, the text is inconsistent.

The berberine and costunolide initials have been unified as capitalized.

Question 13: Page 16, Line 349, please change “of 40 g/ml After” to “of 40 g/ml. After”.

We have fix it. Please see Page 6, Line 151.

Question 14: Page 19, Line 418, “DMEM supplemented with DHW (40μg/ml).” , please unify the font form.

We have revised it as requested by the reviewer. Please see Page 15, Line 426.

Question 15: Page 20, Line 458, please change “(control)or” to “(control) or”.

We have revised it as requested. Please see Page 16, Line 458.

Question 16: Please check some latest published articles and modify the format of the references according the guidelines. Such as “Page 21, Line 470, Please add the issue number.”

We have added the issue number. Please see Page 17, Line 489.

Question 17: Page 21, Line 492, Please check the page number.

We have fixed the page number. Please see Page 18, Line 512.

Question 18: Page 22, Line 523, Please add the page number.

We have added the page number as requested. Please see Page 19, Line 543.

Question 19: Page 22, Line 536, Please check the page number.

We have fixed the page number. Please see Page 19, Line 556.

Question 20: I recommend that the author add a model diagram of the mechanism for readers to better understand.

The model diagram of the mechanism has been added to the article as requested. Please see line 407-410 and Figure 5.

Reviewer #2: The study investigates a Mongolian herbal medicine DaHuangWan in inhibiting hepatoma cell growth through inhibiting EGF signalling. However the following questions should be addressed before acceptance.

1. There are so many signalling pathways related to cancer cell proliferation. Why did the authors target on EGFR signalling?

We did not specifically target EGF signaling. EGF signaling is a key signaling pathway that regulates cell proliferation. When we found that DHW inhibits hepatoma cell growth, we examined multiple signaling pathways including EGF signaling. Both EGFR expression and phosphorylation were significantly increased in hepatoma cells compared to normal hepatocytes (Figure 3A). The proliferation of these HCC cells was inhibited by EGFR inhibitors, and EGFR phosphorylation of these cells was also blocked (Figure 3B). These results confirm the central role of EGF signaling in the proliferation of liver cancer cells, and also lead us to target EGF signaling. As we have discussed in this manuscript: we cannot rule out the possibility that DHW targets other signaling pathways. Previous studies have shown that Berberine prevents HCC from secreting VEGF and down-regulating VEGF mRNA expression (ref. 32). Costunolide also inhibits the proliferation and survival of colorectal cancer cells via inhibiting Wnt/β-Catenin signaling pathway (ref. 33). Although we cannot rule out the possibility of Berberine and Costunolide targeting other signaling pathways, we did prove in this study that in certain tumor cells, like PLC/PRF/5, HT29, and HCC-LM3, DHW inhibit tumor cell proliferation by regulating EGF signaling (lines 389-396)

2. Did the authors consider studying the anti-proliferative effect of other components in DHW? And how the authors exclude the possibility of combined toxicity of costunolide and berberine in the B+C mixture rather than synergistic effect?

We are interested in studying the anti-proliferative effect of other components in DHW. The composition of herbal medicines is very complex and consists of many (perhaps hundreds) ingredients, some of which contribute to its clinical effect. In this study, DHW is a traditional herbal medicine consisting of Coptidis Rhizoma and Aucklandia lappa Decne. It is well known that Berberine and Costunolide are the main bioactive components of Coptidis Rhizoma and Aucklandia lappa Decne. Therefore, our study began to focus on berberine and Costunolide. We do not rule out the possibility that other components in DHW may contribute to the anti-cancer effect of DHW, and are very interested in studying them in future. 

We are careful about the possibility of combined toxicity of Costunolide and Berberine. As shown in Figure 1C, DHW extract inhibited the growth of SMMC-7721, HepG2, Hur7, and LM3 cells in a dose-and time-dependent manner, and these cells have different sensitivities against DHW extract. The effect of DHW on the QSG-7701 human hepatocytes was limited (Fig. 1B), indicating that the inhibition of DHW on the growth of various liver cancer cell lines should not be derived from cytotoxicity (line 217-219). On the other hand, previous studies suggested that the cytotoxicity of Berberine and Costunolide is limited. According to Wang N et al. the LD50 values of Berberine in mice was 713.57 mg/kg and the LD50 values of Berberine in 3T3-L1 cells was 41.76 mg/ml (Wang N et al. J Ethnopharmacol. 2015, 176:35-48). There are few studies on toxicity of Costunolide, 10-50 uM Contunolide is commonly used for cell signal transduction studies through cell culture（e.g. Kim DY et al. Int. J. Mol. Sci. 2019, 20:2926-2947).

3. To confirm the DHW-regulated EGF signalling, it is more reasonable to rescue the EGFR expression rather than blocking it by inhibitor.

Ideally, we wish to rescue the inhibitory effect of DHW on cell proliferation through EGFR overexpression, thereby confirming EGF signaling regulated by DHW. However, this requires that EGFR expression can significantly promote the growth of liver cancer cells. Unfortunately, EGFR homeostasis is tightly controlled in cells. EGFR expression is necessary but not sufficient for cell growth. EGFR overexpression alone is not able to promote the proliferation of hepatoma cell or human hepatocytes such as QSG-7701. Therefore, we cannot perform the ideal experiment described above. We then used an alternative indirect approach: the EGFR inhibitor. Both DHW extract and EGFR inhibitor SC0186 significantly inhibited the growth of liver cancer cells (Figure 3D). However, DHW extract alone and the combination of DHW extract and SC0186 had similar inhibitory effects on the proliferation of PLC/PRF/5 cells (Figure 3D). The combination of the EGFR inhibitor SC0186 with DHW does not bring additional benefits. This shows that DHW extract and SC0186 act on the same signal pathway and the effects cannot be superimposed (line 296-303). Thus, this result indicates that DHW inhibits the proliferation of liver cancer cells by regulating EGF signaling.

4. It is interesting to show that berberine promotes endocytosis while costunolide promotes EGFR ubiquitination. But how did the authors explain for Fig 4D that berberine seems increase the ubiquitin level of EGFR.

Berberine alone did not increase ubiquitin levels in EGFR. As shown in Figure 4D, we quantified the ubiquitin level of EGFR through image J. The Berberine treatment (lane 3) was similar to the control (lane 1) (4.29 vs 3.96). At least three independent experiments confirmed this result and there is no statistically significant difference.

---

## [Decision Letter · Decision Letter 1]

25 Mar 2020

DaHuangWan targets EGF signaling to inhibit the proliferation of hepatoma cells

PONE-D-19-35686R1

Dear Dr. Feng,

We are pleased to inform you that your manuscript has been judged scientifically suitable for publication and will be formally accepted for publication once it complies with all outstanding technical requirements.

With kind regards,

Salvatore V Pizzo

Academic Editor

PLOS ONE

Additional Editor Comments (optional):

Reviewers' comments:

Reviewer's Responses to Questions

**Comments to the Author**

1. If the authors have adequately addressed your comments raised in a previous round of review and you feel that this manuscript is now acceptable for publication, you may indicate that here to bypass the “Comments to the Author” section, enter your conflict of interest statement in the “Confidential to Editor” section, and submit your "Accept" recommendation.

Reviewer #1: All comments have been addressed

2. Is the manuscript technically sound, and do the data support the conclusions?

Reviewer #1: Yes

3. Has the statistical analysis been performed appropriately and rigorously? 

Reviewer #1: Yes

4. Have the authors made all data underlying the findings in their manuscript fully available?

Reviewer #1: Yes

5. Is the manuscript presented in an intelligible fashion and written in standard English?

Reviewer #1: Yes

6. Review Comments to the Author

Reviewer #1: The author's modification to this article is serious and satisfactory. The framework and diagrams of the article have been well modified. I think it is acceptable for publication in this journal.

7. PLOS authors have the option to publish the peer review history of their article (what does this mean?). If published, this will include your full peer review and any attached files.

Reviewer #1: No

---

## [Editor Report · Acceptance letter]

31 Mar 2020

PONE-D-19-35686R1 

DaHuangWan targets EGF signaling to inhibit the proliferation of hepatoma cells 

Dear Dr. Feng:

I am pleased to inform you that your manuscript has been deemed suitable for publication in PLOS ONE. Congratulations! Your manuscript is now with our production department. 

With kind regards,

on behalf of

Dr. Salvatore V Pizzo 

Academic Editor

PLOS ONE